# Polymorphic cobalt diselenide as extremely stable electrocatalyst in acidic media via a phase-mixing strategy

Xiao-Long Zhang[1,6], Shao-Jin Hu[2,6], Ya-Rong Zheng [1,6], Rui Wu[1], Fei-Yue Gao[1], Peng-Peng Yang[1], Zhuang-Zhuang Niu[1], Chao Gu[1], Xingxing Yu[1], Xu-Sheng Zheng[3], Cheng Ma[4], Xiao Zheng [2], Jun-Fa Zhu [3], Min-Rui Gao [1]* & Shu-Hong Yu [1,5]*

Many platinum group metal-free inorganic catalysts have demonstrated high intrinsic activity for diverse important electrode reactions, but their practical use often suffers from undesirable structural degradation and hence poor stability, especially in acidic media. We report here an alkali-heating synthesis to achieve phase-mixed cobalt diselenide material with nearly homogeneous distribution of cubic and orthorhombic phases. Using water electroreduction as a model reaction, we observe that the phase-mixed cobalt diselenide reaches the current density of 10 milliamperes per square centimeter at overpotential of mere 124 millivolts in acidic electrolyte. The catalyst shows no sign of deactivation after more than 400 h of continuous operation and the polarization curve is well retained after 50,000 potential cycles. Experimental and computational investigations uncover a boosted covalency between Co and Se atoms resulting from the phase mixture, which substantially enhances the lattice robustness and thereby the material stability. The findings provide promising design strategy for long-lived catalysts in acid through crystal phase engineering.

[1] Division of Nanomaterials & Chemistry, Hefei National Laboratory for Physical Sciences at the Microscale, CAS Center for Excellence in Nanoscience, Hefei Science Center of CAS, Collaborative Innovation Center of Suzhou Nano Science and Technology, Department of Chemistry, University of Science and Technology of China, 230026 Hefei, China. [2] Division of Theoretical and Computational Sciences, Hefei National Laboratory for Physical Sciences at the Microscale, Department of Chemical Physics, University of Science and Technology of China, 230026 Hefei, Anhui, P. R. China. [3] National Synchrotron Radiation Laboratory, University of Science and Technology of China, 230029 Hefei, P. R. China. [4] Department of Materials Science and Engineering, University of Science and Technology of China, 230026 Hefei, China. [5] Dalian National Laboratory for Clean Energy, 116023 Dalian, China. [6] These authors contributed equally: Xiao-Long Zhang, Shao-Jin Hu, Ya-Rong Zheng. *email: mgao@ustc.edu.cn; shyu@ustc.edu.cn

Clean-energy technologies such as fuel cells and electrolyzers require even more active and stable electrocatalysts that accelerate the multi-proton/multi-electron-involved electrode reactions[1,2]. Previous studies have shown diverse approaches to boosting catalytic activity, such as chemical doping[3,4], strain[5,6] and defect engineering[7], alloying[8–10] and dealloying[11]. Moreover, several activity descriptors that guide the design of energetic catalysts—including the $d$-band center associated with oxygen reduction reaction (ORR)[12] and the $e_g$ filling associated with oxygen evolution reaction (OER)[13], as well as the $\triangle G_H$ binding energy related with hydrogen evolution reaction (HER)[14]—have been proposed, advancing the development of new electrocatalysts. However, the degradation of many active catalysts in harsh electrochemical environments, such as $RuO_2$ in OER process[15] and Pt alloys in ORR process[10], has remained a major challenge that limits the device efficiency and cost effectiveness.

The stability of electrocatalysts could be governed by multi-parameters, including the bulk Pourbaix thermodynamics, dissolution potential, and solution pH. In practical electrochemical process, catalysts may suffer from poisoning by impurity gases (e.g., CO)[16] and corrosion by in situ formed radicals (e.g., ·OH)[17], as well as particle coarsening[18], which cause severe deactivation. Prior efforts suggest that the stability can be improved in various ways—for example, by catalyst encapsulation[19,20], compositional modulation[10,21,22], or coupling with catalyst supports[23]. Despite improvements in catalyst stability with these strategies, most operational stability studies have been performed on Pt-group catalysts[10,20–23]. As with emerging nonprecious metal-based catalysts, their long-term stability, particularly in acidic electrolyte, is unsatisfactory. This consequently leads to the problem of using nonprecious catalysts in polymer electrolyte membrane-based electrochemical devices, which need low pH environment for operation.

Here we describe a way to improve the stability of polymorphic cobalt diselenide ($CoSe_2$) catalyst through homogeneous phase mixing between cubic and orthorhombic phases. In acidic electrolyte (0.5 M $H_2SO_4$, pH = 0), the phase-mixed $CoSe_2$ (m-$CoSe_2$) requires an overpotential of 124 mV at 10 mA cm$^{-2}$, which was perfectly retained for over 400 h of continuous operation, consistent with the observation from accelerated stability tests after 50,000 electrochemical cycles. This remarkable HER stability of phase-mixed $CoSe_2$ at low pH can be ascribed to the greater covalency between Co and Se atoms, resulting in lattice configuration with enhanced robustness. We expect that such phase mixing method could be applicable to other polymorphic materials for designing new electrocatalysts with better catalytic performances.

## Results

### Synthesis and characterization of m-CoSe$_2$.
Ultrathin cubic $CoSe_2$ (c-$CoSe_2$) nanobelts were synthesized as precursors using the method described previously[24] (Supplementary Fig. 1). We recently reported on the complete structural phase transition of c-$CoSe_2$ to orthorhombic $CoSe_2$ (o-$CoSe_2$) induced by phosphorus doping at a high temperature of 400 °C (ref. [25]). In this study, we develop an alkali-heating approach to produce m-$CoSe_2$ with nearly homogeneous distribution of cubic and orthorhombic phases (Supplementary Fig. 2). Briefly, the as-made c-$CoSe_2$ powder was placed in 5 M KOH solution and heated at 200 °C for 12 h. This harsh condition causes the leaching of Co and Se at defective sites of c-$CoSe_2$, leaving atomic vacancies (Fig. 1a). Buseck et al.[26] proposed that lattice defects in cubic pyrites could offer sites to drive its transition to orthorhombic marcasite. Because pyrite $CoSe_2$ (100) shows almost the same atomic

arrangement with marcasite $CoSe_2$ (101) (Supplementary Fig. 3), it permits locally epitaxial growth of o-$CoSe_2$ at these vacancy defects. Density functional theory (DFT) calculations in Fig. 1b reveal that the c-$CoSe_2$ (100)‖o-$CoSe_2$ (101) interfacial energy is smaller than the surface energies of c-$CoSe_2$ (100) and o-$CoSe_2$ (101), suggesting that the epitaxial growth of o-$CoSe_2$ is thermodynamically favored. We thus conclude that the in situ formed atomic vacancies mediated the formation of o-$CoSe_2$ phase, resulting in unusual m-$CoSe_2$ (Fig. 1a), which is analogous to the Se vacancies induced 1T-to-1H transition in $PtSe_2$ film reported recently[27].

Our series of control experiments show that alkali and temperature are critical to trigger structural phase transition in c-$CoSe_2$ (Supplementary Figs. 4–13). Without alkali or aging c-$CoSe_2$ in 5 M KOH at room temperature, the Co and Se leaching from c-$CoSe_2$ was kinetically limited, as confirmed by inductively coupled plasma atomic emission spectroscopy (ICP-AES; Fig. 1c). Consequently, it lacks the needed vacancy defects in c-$CoSe_2$ where the phase transition occurs. The above experiments also uncover that heating c-$CoSe_2$ at 200 °C for 12 h in 5 M KOH would result in the optimal m-$CoSe_2$ (Supplementary Figs. 8–13). Alkali-heating treatment has been widely applied on carbon materials[28] (e.g., graphene[29] and carbon nanotubes[30]) to improve the porosity, electrochemical performance, and corrosion resistance. We note that no collapse of c-$CoSe_2$ structure under above harsh conditions is unexpected, which implies the remarkable robustness of the achieved m-$CoSe_2$.

Scanning transmission electron microscopy (STEM) image in Fig. 2a shows surface scratches on m-$CoSe_2$ after the alkali-heating process, consistent with the observations from scanning electron microscopy (SEM; Supplementary Fig. 14a) and transmission electron microscopy (TEM; Supplementary Fig. 14b) images. Atomic force microscopy (Supplementary Fig. 15a) and high-angle annular dark field (HAADF; Supplementary Fig. 15b) images clearly reveal that the alkali-heating treatment etches c-$CoSe_2$ precursors and has created slit nanopores with sizes ranging from 1.4 to 7.7 nm (Supplementary Fig. 16). These nanopores were formed owing to the Co and Se leaching, which makes atomic level interfaces with high surface free energy where the rearrangement of Co and Se could happen. Energy-dispersive X-ray spectrum (EDX) elemental mapping in Fig. 2b shows a uniform spatial distribution of Co and Se in m-$CoSe_2$ sample, and its overall Co:Se ratio remains about 1:2 based on EDX analysis (Supplementary Fig. 17).

Figure 2c shows the inverse fast Fourier transform (FFT) image of m-$CoSe_2$ from a typical HAADF-STEM image (Supplementary Fig. 18a), which clearly displays the distribution the c-$CoSe_2$ (green) and o-$CoSe_2$ (red). Inverse FFT images from other regions of m-$CoSe_2$ nanobelts further confirm the unique phase-mixed microstructure (Supplementary Fig. 19). Atomic-resolution HAADF-STEM image in Fig. 2d reveals that the c-$CoSe_2$ and o-$CoSe_2$ phases can be distinguished from their different crystalline structures, where the FFT pattern (inset in Fig. 2d) can be interpreted as two sets of patterns from the [111] zone axis of c-$CoSe_2$ and [100] zone axis of o-$CoSe_2$. Figure 2e, f shows HAADF-STEM images taken at higher magnifications from the areas marked by the red and yellow dashed squares. The c-$CoSe_2$ lattice exhibits a honeycomb structure (Fig. 2e), which is unambiguously discriminated from the o-$CoSe_2$ with parallel line lattices (Fig. 2f). The corresponding FFT patterns demonstrate cubic and orthorhombic phases at the two regions (Insets of Fig. 2e, f), consistent with our FFT simulation results (Supplementary Fig. 20). The co-existed two phases are further visualized by high-resolution TEM images taken at different areas of the m-$CoSe_2$, as shown in Supplementary Fig. 21. X-ray diffraction (XRD; Fig. 2g) studies also support the phase transition of initial

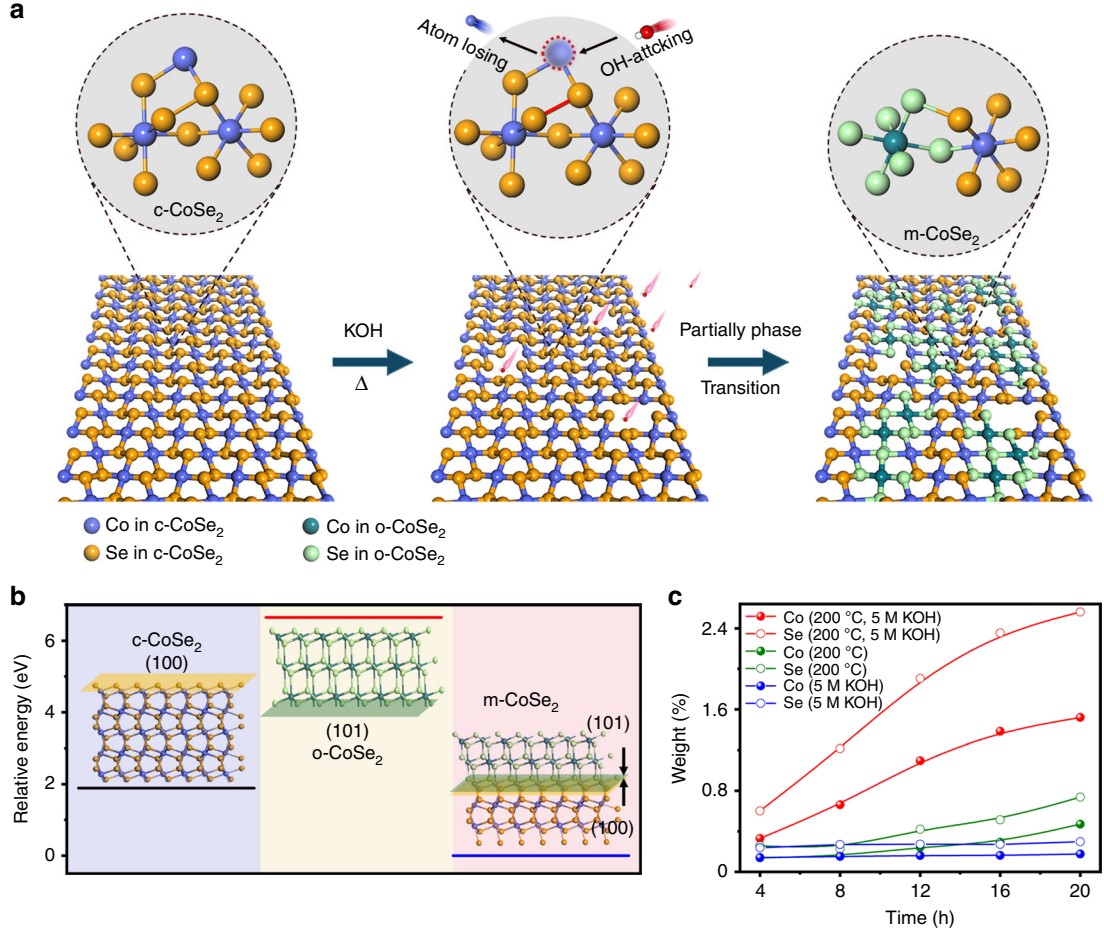

**Fig. 1** Structural phase transition in c-CoSe₂. **a** Schematic illustration of the defects-mediated structural phase transition from c-CoSe₂ to m-CoSe₂ through an alkali-heating approach. **b** Surface/interfacial energy diagram of c-CoSe₂ (100), o-CoSe₂ (101), and m-CoSe₂ (100||101), respectively. Insets in **b** show the corresponding crystal structures. **c** ICP-AES results compare the Co and Se leaching from c-CoSe₂ at different reaction conditions, revealing that KOH and temperature are critical to drive the structural phase transition.

c-CoSe₂ to phase-mixed CoSe₂ (also see Supplementary Figs. 8a, 10a and 12a).

Using Raman spectroscopy, we tracked the structural phase evolution in c-CoSe₂ (Supplementary Figs. 8b, 10b and 12b). The Raman active peak at $189\,cm^{-1}$ for c-CoSe₂ originates from the Se–Se stretching mode[31]. With tuning temperature, time or KOH concentration, a new Raman peak at $167\,cm^{-1}$ arises gradually, indicative of the formation of orthorhombic phase. Nevertheless, Fig. 2h reveals that this new peak has $\sim 7\,cm^{-1}$ leftward shift compared with that of as-synthesized pure o-CoSe₂ ($174\,cm^{-1}$), suggesting the strong electronic phase interaction in m-CoSe₂ (ref. [32]). We also studied the work function of c-, o-, and m-CoSe₂ by ultraviolet photoelectron spectroscopy (UPS; Fig. 2i). Our measurements display a lower work function of m-CoSe₂ (3.7 eV) compared with c-CoSe₂ (3.9 eV) and o-CoSe₂ (4.6 eV), which hint at the optimized electronic structures of m-CoSe₂ that enable a faster electron transfer and hence superior catalytic capability.

**HER activity and stability on m-CoSe₂ catalysts**. We evaluated the electrocatalytic activities of c-, o-, and m-CoSe₂ toward HER in Ar-saturated 0.5 M H₂SO₄ electrolyte with Ohmic drop correction (Supplementary Fig. 22). Rotating disk electrode (RDE) testing shows that m-CoSe₂ requires an overpotential of mere 124 mV at 10 mA cm⁻², far exceeding that of 226 mV for c-CoSe₂ and 273 mV for o-CoSe₂ (Fig. 3a–c, Supplementary Fig. 22a). Tafel analysis gives a slope of 60, 70, and $37\,mV\,dec^{-1}$ (here

"dec" means decade, or one order of magnitude) for c-, o-, and m-CoSe₂ catalysts (Fig. 3b), respectively. The lower Tafel slope of $37\,mV\,dec^{-1}$ for m-CoSe₂ indicates its HER superiority as compared to c- and o-CoSe₂ catalysts. Further, the Tafel slope of $37\,mV\,dec^{-1}$ obtained for acidic HER also suggests a Tafel-step-determined Volmer-Tafel pathway that likely works in the m-CoSe₂ catalyst[33]. H₂ oxidation currents from the rotating ring-disk electrode measurements (Pt ring at 0.5 V versus the reversible hydrogen electrode (RHE)) confirm the selective H₂ production on the above catalysts (Supplementary Fig. 23). The HER performances of the m-CoSe₂ catalyst with different reaction temperatures, times, and KOH concentrations were systematically studied, as shown in Supplementary Figs. 24–26. We note that the HER kinetic metrics (e.g., 124 mV overpotential at 10 mA cm⁻² and Tafel slope of $37\,mV\,dec^{-1}$) gained from the RDE testing ranks among the superb for noble-metal-free catalysts documented in acid[34,35]. The HER activity in acid of m-CoSe₂ also performs better than CoSe₂ catalysts with various phases reported previously (Supplementary Fig. 27). We further detected the amount of generated H₂ gas via gas chromatography, which is well consistent with the theoretical value, corresponding to a Faradaic efficiency of 99.6% (Supplementary Fig. 28). The marked HER activity of m-CoSe₂ can be ascribed to the large number of nanopores that offer greater accessibility to active sites, as well as the superior electronic properties originated from the unique phase-mixed structure.

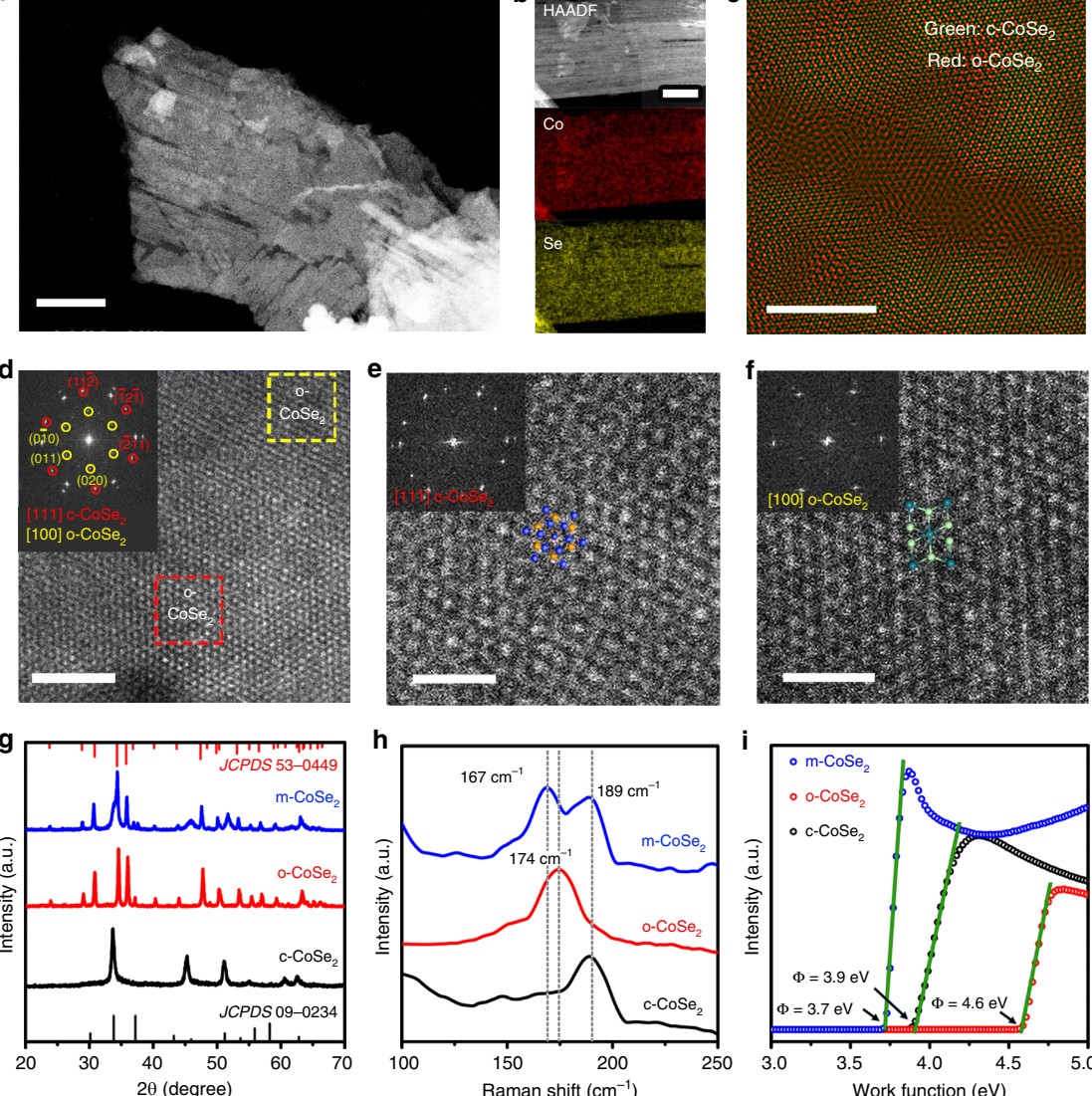

**Fig. 2** Physical characterization of m-CoSe$_2$. **a** STEM image of a typical m-CoSe$_2$ nanobelt. Scale bar, 100 nm. **b** STEM-EDX elemental mapping of m-CoSe$_2$, showing the homogeneous distribution of Co (red) and Se (yellow), respectively. Scale bar, 50 nm. **c** Atomic-resolution image reconstructed by overlapping the inverse FFT patterns shown in Supplementary Fig. 18c, d. Scale bar, 5 nm. **d** Atomic-resolution HAADF-STEM image of m-CoSe$_2$. The regions marked by the red and yellow dashed squares indicate c-CoSe$_2$ and o-CoSe$_2$, respectively. Inset in **d** gives corresponding FFT pattern, exhibiting two sets of patterns from the [111] zone axis of c-CoSe$_2$ and [100] zone axis of o-CoSe$_2$. **e, f** High-magnification HAADF-STEM images taken from the red and yellow dashed squares in **d**, respectively. Scale bars, 1 nm. Insets in **e** and **f** show the corresponding FFT patterns, evidencing the c-CoSe$_2$ and o-CoSe$_2$ phases, respectively. **g–i** XRD patterns (**g**), Raman spectra (**h**), and UPS spectra (**i**) of the c-, o-, and m-CoSe$_2$.

The m-CoSe$_2$ catalyst that survives the harsh alkali-heating conditions prompted us to carry out a comprehensive assessment of its long-term stability in acidic solution. Figure 3a–c displays the HER polarization curves of m-, c-, and o-CoSe$_2$ catalysts after different numbers of potential cycles between −0.3 and 0 V versus RHE, respectively. We see that the m-CoSe$_2$ catalyst requires 124 mV overpotential at 10 mA cm$^{-2}$; this value was nearly retained after 50,000 potential cycles (Fig. 3a). By comparison, c-CoSe$_2$ and o-CoSe$_2$ catalysts require 63 mV (Fig. 3b) and 45 mV (Fig. 3c) additional overpotentials at 10 mA cm$^{-2}$ after mere 2000 cycles. We also performed electrochemical impedance spectroscopy (EIS) at 200 mV overpotential to probe the charge transfer resistance ($R_{ct}$) that normalized by electrochemically active surface area (ECSA) for these catalysts. Figure 3e reveals that the $R_{ct}$ of m-CoSe$_2$ is 1524 versus ~13,660 ohm cm$_{ECSA}^2$ for c-CoSe$_2$ (Fig. 3f) and ~8100

ohm cm$_{ECSA}^2$ for o-CoSe$_2$ (Fig. 3g). The smallest $R_{ct}$ of m-CoSe$_2$ suggests its superior charge transfer kinetics, agreeing with UPS measurements. Figure 3e and Supplementary Fig. 29 also reveal that no appreciable change in $R_{ct}$ was observed for m-CoSe$_2$ after 50,000 cycles, whereas the $R_{ct}$ values substantially increased to 97,790 ohm cm$_{ECSA}^2$ for c-CoSe$_2$ (Fig. 3f) and 22,651 ohm cm$_{ECSA}^2$ for o-CoSe$_2$ (Fig. 3g) after 2000 cycles, consistent with results shown in Fig. 3a–c. The resistance increase hints at structural degradation of c-CoSe$_2$ and o-CoSe$_2$ in acid. Using chronopotentiometry (Fig. 3h), we found that almost no additional overpotential was required to maintain the current density of 10 mA cm$^{-2}$ over 400 h of continuous operation in 0.5 M H$_2$SO$_4$, consistent with chronoamperometric measurement (Supplementary Fig. 30). All these experiments clearly demonstrate the excellent long-term catalytic stability of phase-mixed CoSe$_2$ in acidic electrolyte. We further highlight that m-CoSe$_2$

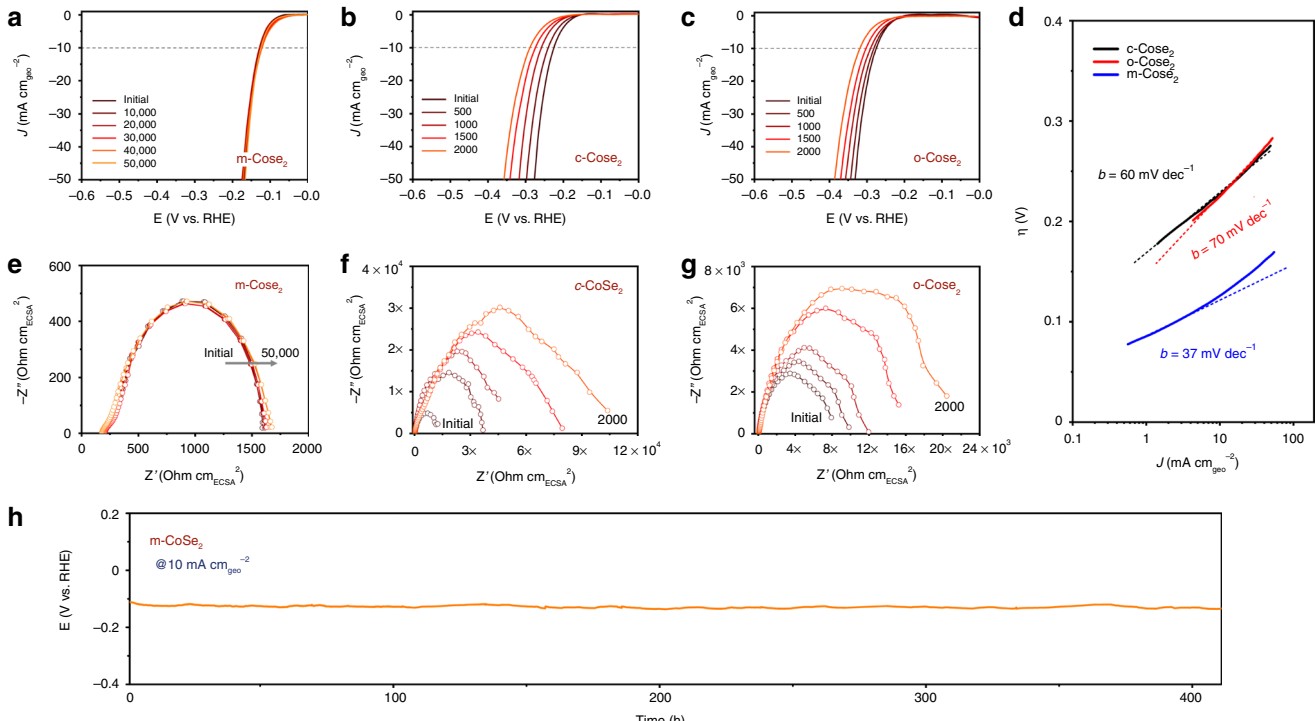

**Fig. 3** Electrochemical stability of m-, c-, and o-CoSe$_2$. **a–c** HER polarization curves of m-, c-, and o-CoSe$_2$ catalysts before and after different potential cycles, respectively. Catalyst loading: ~1.02 mg cm$^{-2}$. Sweep rate: 2 mV s$^{-1}$. **d** Tafel plots for m-, c-, and o-CoSe$_2$ catalysts derived from their initial polarization curves. **e–g** EIS Nyquist plots of the m-, c-, and o-CoSe$_2$ catalysts before and after different potential cycles, respectively. **h** Chronopotentiometry ($E \sim t$) recorded on m-CoSe$_2$ catalyst at the constant current density of 10 mA cm$^{-2}$, exhibiting the exceptional long-term stability.

catalyst can retain the HER activity even storing in laboratory for 10 months, showing its outstanding environmental stability (Supplementary Fig. 31).

**Stability study of m-CoSe$_2$.** We next combined multiple characterization techniques to examine the structural stability of the studied catalysts. Figure 4a shows selective-area electron diffraction patterns of the m-CoSe$_2$ catalyst before and after 10,000, 30,000, and 50,000 potential cycles along the same zone axis. Careful analysis of these diffraction spots uncovered two sets of patterns belonging to c-CoSe$_2$ (red cycles) and o-CoSe$_2$ (yellow cycles). Notably, these diffraction peaks almost remain unchanged even after 50,000 cycles, in excellent agreement with the XRD (Fig. 4b) and Raman (Fig. 4c) results. Post-mortem TEM analysis reveals that the belt-like morphology of pristine m-CoSe$_2$ was well retained after cycling tests (Supplementary Fig. 32). X-ray photoelectron spectra (XPS) measurements reveal Co–O (781.09 eV)[36] and Se–O (58.59 eV)[37] bonds for the initial m-CoSe$_2$ catalyst, which were completely removed during the durability tests (Fig. 4d, e). Furthermore, we observed that the valence states of both Co and Se were well maintained after 50,000 cycles, suggesting that the electronic structure of m-CoSe$_2$ was not perturbed (Fig. 4d, e, Supplementary Fig. 33). By contrast, Raman spectroscopy studies demonstrate that a noticeable peak at 248 cm$^{-1}$ arises for c-CoSe$_2$ and o-CoSe$_2$ catalysts after mere 500 cycles (Supplementary Fig. 34). We attribute this new Raman active peak to the formation of amorphous Se (ref.[38]). Our ICP-AES analyses in Fig. 4f present severe Co and Se leaching into the electrolyte for c-CoSe$_2$ and o-CoSe$_2$, whereas the m-CoSe$_2$ catalyst almost remains physically intact (also see photographs in Supplementary Fig. 35). Additionally, we added the cycled electrolytes into the sodium acetate and nitroso-R salt solution, where the green

solution turned to red owing to the formation of Co[C$_{10}$H$_4$ONO (SO$_3$Na)$_2$]$_3$ complex[39] (Fig. 4g), again showing the electrochemical leaching of Co for c-CoSe$_2$ and o-CoSe$_2$ samples. Together, the above results unambiguously illustrate the excellent catalytic and structural stability of the new phase-mixed CoSe$_2$ catalyst in acidic solutions.

**DFT calculation and enhancement mechanism.** We now turn to discuss the structural and chemical features that potentially affect and govern the remarkable stability of m-CoSe$_2$ catalyst. We remark that cubic pyrite and orthorhombic marcasite CoSe$_2$ are polymorphs; they both possess Co$^{2+}$ occurring in octahedral coordination and contain the characteristic Se–Se pairs[40] (Fig. 5a; Supplementary Fig. 36). However in cubic pyrite CoSe$_2$, the octahedra share corners, whereas in orthorhombic marcasite they share edges[41] (Fig. 5a). Structurally, rotating half of Se–Se pairs in cubic pyrite through 90° can yield the orthorhombic marcasite structure[40]. The perfect lattice similarity between pyrite {001} and marcasite {101} allows for the epitaxial growth of o-CoSe$_2$ on c-CoSe$_2$ (Fig. 5b, inset), as confirmed by DFT calculations[41,42]. Although theoretically feasible, no nanostructures with a homogeneous mixture of pyrite and marcasite have been synthesized before. Here the integration of o-CoSe$_2$ into c-CoSe$_2$ via harsh alkali-heating process successfully produces m-CoSe$_2$ structure, which not only brings rigid material lattices but also creates electronic structure perturbations that permit greater covalent bonding forces between Co and Se. Therefore, the exceptional robustness of m-CoSe$_2$ in acidic environments is expected.

To better understand the origin of the notable stability, X-ray absorption near-edge structure (XANES) spectroscopy, and DFT calculations were conducted. Figure 5c gives the XANES spectra of c-CoSe$_2$ and m-CoSe$_2$ at Co K-edge, which both show the pre-

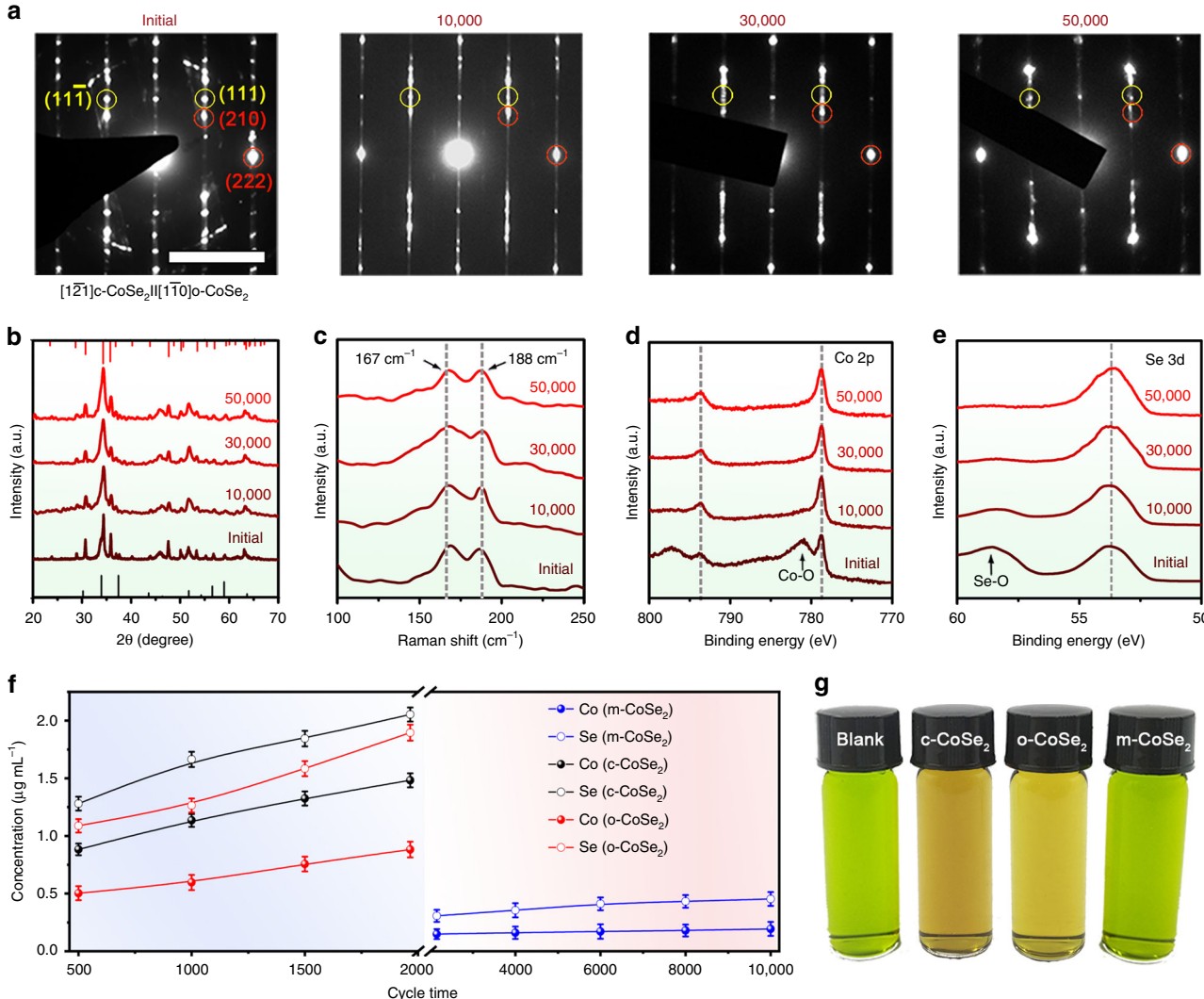

**Fig. 4** Structural stability of m-CoSe₂. **a–e,** SAED patterns (**a**), XRD patterns (**b**), Raman spectra (**c**), Co 2p XPS spectra (**d**), and Se 3d XPS spectra (**e**) of the m-CoSe₂ catalyst before and after different potential cycles. Scale bar in **a**, 5 1/nm. **f** ICP-AES measurements compare the Co and Se dissolved into acidic electrolyte after different potential cycles for m-, c-, and o-CoSe₂ catalysts. **g** Colorimetric comparison of the cycled electrolytes added in sodium acetate and nitroso-R salt solution. No color change was seen for m-CoSe₂, indicating its marked structural stability in acidic electrolyte. Nitroso-R salt was used as the color indicator. The freshly-made c- and o-CoSe₂ catalysts were treated in pure deionized water at 200 °C for 12 h before cycling to remove any potential unstable or amorphous components.

edge feature that indicates Co ions in the octahedral environment[43]. The shoulder feature at ~7717 eV originates from the covalent nature of the Co-Se bond, analogous to previous observations in other systems[39,44]. The m-CoSe₂ catalyst reveals a considerable shift of the Co K-edge to a higher energy versus c-CoSe₂ (Fig. 5c), in agreement with XPS and Co L-edge measurements (Supplementary Figs. 33 and 37). This suggests a decreased occupancy of the antibonding $e_g^*$ orbitals, pointing to a stronger Co 3d–Se 4p orbital hybridization. Our Se K-edge XANES spectra shown in Fig. 5d reveal that m-CoSe₂ exhibits a stronger peak at 12661 eV versus c-CoSe₂, also verifying a more covalent Co–Se bonding orbital system[45]. As a result, we reason that the new phase-mixed CoSe₂ catalyst shows a decreased occupancy of the antibonding $e_g^*$ orbitals of Co 3d, which tends to attract more ligand 4p orbitals, enabling a greater covalency of Co–Se bond and thus substantially enhanced stability (Fig. 5e, f).

We further created the m-CoSe₂ DFT model by combining the c-CoSe₂ (100) and o-CoSe₂ (101) surfaces ended as [Se-Co-Se], which give nonpolar terminations[46] (Fig. 5b, Supplementary Figs. 38 and 39). DFT calculations show that epitaxial growth of

o-CoSe₂ (101) on c-CoSe₂ (100) results in a significantly reduced free energy as compared with original c-CoSe₂ (100), suggesting that the formation of m-CoSe₂ is thermodynamically favorable (Fig. 5g). This accordingly implies the superior phase stability of m-CoSe₂. Moreover, the density of states (DOS) results reveal that the m-CoSe₂ catalyst exhibits lower states in the characteristic low-DOS region close to the Fermi level (Fig. 5h). The less electronic states near the Fermi level again indicate the stability of m-CoSe₂. Our calculations also demonstrate that the epitaxial growth of o-CoSe₂ on c-CoSe₂ is limited to a few [Se-Co-Se] layers, while further growth causes structure instability (Supplementary Figs. 40–44), agreeing with the well-distributed phase mixture of m-CoSe₂ we have achieved.

## Discussion
A major challenge of adopting noble-metal-free catalysts in polymer electrolyte membrane-based electrochemical devices is to retain their stability in acidic environments. We have shown here that m-CoSe₂ is extremely stable for catalyzing HER in acid,

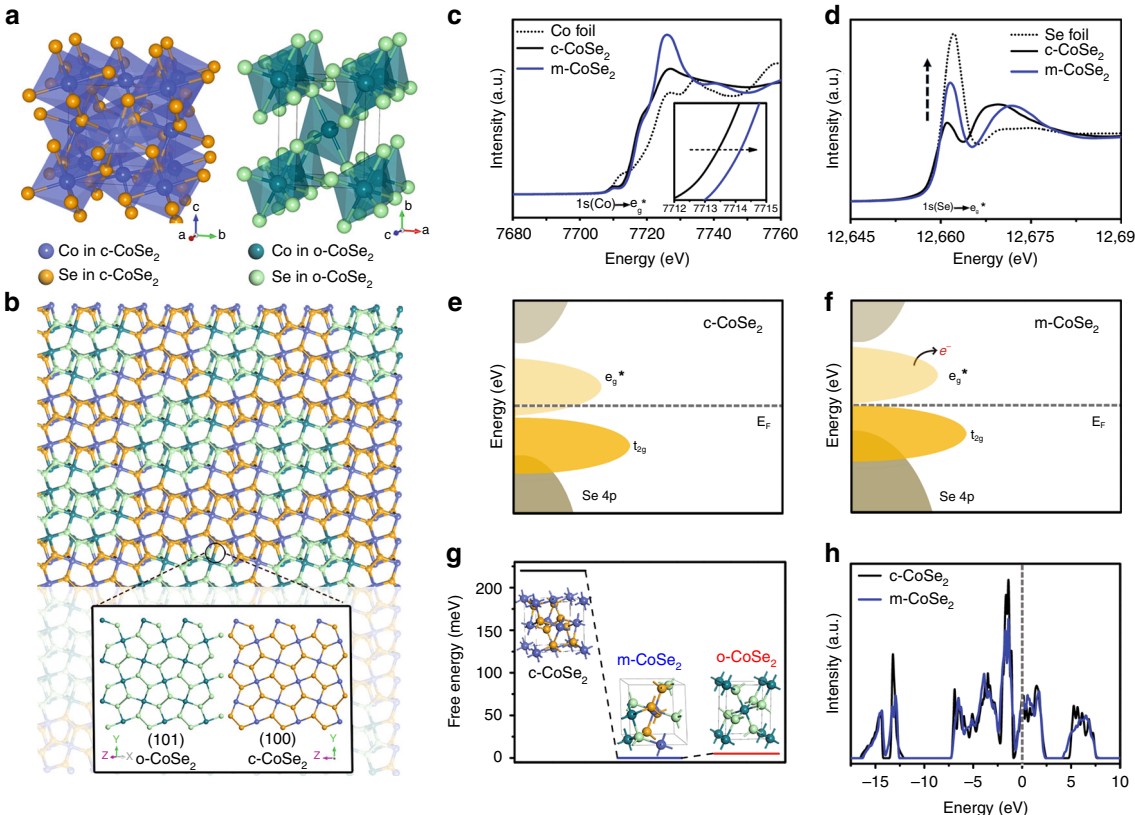

**Fig. 5** DFT calculation and stability mechanism. **a** Crystal structures of the c-CoSe$_2$ (left) and o-CoSe$_2$ (right). **b** Schematic structure model of m-CoSe$_2$ showing the [c-CoSe$_2$ (100)||o-CoSe$_2$ (101)] interface. Insets are the atomic models of the o-CoSe$_2$ (101) and c-CoSe$_2$ (100) surfaces, respectively. **c**, **d** XANES spectra recorded at the Co K-edge (**c**) and Se K-edge (**d**) of the Co foil, Se foil, c-CoSe$_2$, and m-CoSe$_2$, respectively. Inset in **c** shows the zoomed view of the Co K-edge spectra. **e**, **f** Schematic rigid band diagrams of the c-CoSe$_2$ (**e**) and m-CoSe$_2$ (**f**). **g** Calculated free energy of the c-, o-, and m-CoSe$_2$ unit cells. Insets show corresponding unit cell structures. **h** DOS of the c-CoSe$_2$ and m-CoSe$_2$ with the Fermi level aligned at 0 eV.

whereas the stabilities of both c-CoSe$_2$ and o-CoSe$_2$ are ordinary. The tremendously enhanced stability of m-CoSe$_2$ can be explained by its robust lattice and the greater covalency of Co-Se bonds after phase mixing, as evidenced by multiple characterizations and DFT calculations. Given many polymorphic material systems exist, we expect that such phase-mixed engineering methodology could be intensively extended for designing cost-effective and better-performing catalysts used in acid, thus aiding the advancement of polymer electrolyte fuel cells and electrolyzers.

## Methods

**Material synthesis**. All chemicals were used as received without further purification. The m-CoSe$_2$ was synthesized through a two-step method. First, c-CoSe$_2$ nanobelts were synthesized as described in our previous works[24]. Next, 25 mg c-CoSe$_2$ nanobelts was dispersed in 2.5 ml 5 M KOH solution. After drastically stirring at room temperature for 30 min, the black solution was transferred into a Teflon-lined autoclave (25 ml), which was sealed and heated at 200 °C for 12 h. After reaction, the obtained m-CoSe$_2$ powder was carefully washed and dried before use.

The o-CoSe$_2$ nanobelts were synthesized by a method developed previously[47]. Briefly, 0.140 g CoSO$_4$·7H$_2$O, 0.079 g Se, and 0.180 g C$_{18}$H$_{33}$NaO$_2$ were added into a mixed solution (40 mL) with a volume ratio of $V_{DETA}/V_{DIW}$ = 2:1 (DIW = deionized water). After stirring for 30 min, the mixture was transferred into a Teflon-lined autoclave and heated at 180 °C for 30 h. The final product was carefully washed and dried before use.

**Material characterizations**. The achieved samples were examined by multiple analytic techniques. XRD was taken with a Philips X'Pert Pro Super X-ray diffractometer with Cu Kα radiation ($\lambda$ = 1.54178 Å). The morphology of the samples was investigated by SEM (Zersss Supra 40) and TEM (Hitachi H7650). The STEM and HRTEM images, SAED, and EDX elemental mappings were taken on a JEMARM 200F Atomic Resolution Analytical Microscope with an acceleration

voltage of 200 kV. Raman spectra were measured on a Raman microscope (Renishaw®) excited with a 514 nm excitation laser. ICP-AES data were obtained by an Optima 7300 DV instrument. N$_2$ adsorption/desorption isotherms were recorded on an ASAP 2020 accelerated surface area and a porosimetry instrument (Mictromeritics), equipped with an automated surface area, at 77 K using Barrett–Emmett–Teller calculations. Ultraviolet photoelectron spectroscopy was carried out at the BL11U beamline of National Synchrotron Radiation Laboratory in Hefei, China. The X-ray absorption spectra of Co and Se K-edges were obtained at the beamline 14W1 of Shanghai synchrotron Radiation Laboratory (China). XPS was taken on an X-ray photoelectron spectrometer (ESCALab MKII) with an X-ray source (Mg Kα $h\nu$ = 1253.6 eV). The X-ray absorption spectra of Co L-edges were performed on the BL10B beamline of National Synchrotron Radiation Laboratory in Hefei (China).

**Electrochemical measurements**. All the electrochemical measurements were measured in a conventional three-electrode cell at ambient temperature connected to a Multipotentiostat (IM6ex, ZahnerElectrik, Germany). Ag/AgCl (3.5 M KCl) electrode and graphite rod were used as the reference and counter electrodes, respectively. The potentials reported in this work were normalized versus the RHE through a standard RHE calibration described elsewhere[48]. A RDE with glassy carbon (PINE, 5.00 mm diameter, disk area: 0.196 cm$^2$) was used as the working electrode.

To make the working electrodes, 5 mg catalyst powder was dispersed in 1 mL of 1:3 v/v isopropanol/DIW mixture with 40 μL Nafion solution (5 wt%), which was ultrasonicated to yield a homogeneous ink. Then, 40 μL catalyst ink was pipetted onto the glassy carbon substrate (catalyst loading: ~1.02 mg cm$^{-2}$). HER measurements were conducted in 0.5 M H$_2$SO$_4$. The fresh electrolytes were bubbled with pure argon for 30 min before measurements. The polarization curves were obtained by sweeping the potential from −0.55 to 0.05 V versus RHE with a sweep rate of 2 mV s$^{-1}$ and 1600 r.p.m. (to remove the H$_2$ bubbles formed in situ) at ambient temperature. The EIS measurement was performed in the same configuration at 200 mV overpotential over a frequency range from 100 KHz to 100 mHz at the amplitude of the sinusoidal voltage of 5 mV. The polarization curves were re-plotted as overpotential ($\eta$) versus log current (log $j$) to get Tafel plots to assess the HER kinetics of investigated catalysts. The Tafel slope ($b$) can be

obtained by fitting the linear portion of the Tafel plots with the following equation:

$$\eta = b \log(j) + a. \quad (1)$$

The m-CoSe$_2$-modified carbon fiber paper (catalyst loading: ~1.02 mg cm$^{-2}$) was used as a working electrode to perform chronopotentiometry experiments at a constant current density of 10 mA cm$^{-2}$. The accelerated stability measurements were performed by potential cycling between −0.3 and 0 V versus RHE with a sweep rate of 200 mV s$^{-1}$. After cycling, the resultant electrode was used for polarization curves with a sweep rate of 2 mV s$^{-1}$. To estimate the double-layer capacitance, cyclic voltammograms were performed at different sweep rates in the potential region of 0.0–0.1 versus RHE at ambient temperature. All the polarization curves were corrected with iR compensation that resulted from the solution resistance. We employed the ICP-AES method to analyze the etching rate of m-CoSe$_2$ during chronopotentiometry experiments in 0.5 M H$_2$SO$_4$. The catalyst was loaded on the $1 \times 1$ cm$^2$ carbon paper substrate (~1.02 mg cm$^{-2}$). Each ICP-AES data point was collected for three times. The ECSA of the catalyst is calculated from the double-layer capacitance based on the equation:

$$\mathrm{ECSA} = \frac{C_{\mathrm{dl}}}{C_{\mathrm{s}}}, \quad (2)$$

where $C_{\mathrm{s}}$ is the specific capacitance of the catalyst or the capacitance of an atomically smooth planar surface of the material per unit area under identical electrolyte conditions. And a general specific capacitance of $C_{\mathrm{s}} = 0.035$ mF cm$^{-2}$ is adopted based on typical reported value[49].

**Colorimetric comparison measurements**. The m-CoSe$_2$-modified carbon fiber paper (catalyst loading: ~5.10 mg cm$^{-2}$) was used as a working electrode to perform accelerated stability experiments. And the electrolyte was collected after 50,000 electrochemical cycles. Then, 5 mL electrolyte, 5 mL sodium acetate solution (250 g L$^{-1}$), and 3 mL of nitroso-R salt solution (10 g L$^{-1}$) were added into a 50 mL beaker. The Co$^{2+}$ can form a red-colored complex (Co[C$_{10}$H$_4$ONO(SO$_3$Na)$_2$]$_3$) with the nitroso-R salt in acetate buffer solution.

**DFT calculations**. The DFT calculations were performed using the plane-wave code Vienna ab-initio simulation package (VASP)[50] program with the projector augmented wave (PAW)[51] method. The convergence criterion of the electronic self-consistent iteration was set to be $10^{-6}$ eV and the kinetic energy cutoff is 400 eV. The atomic positions were relaxed until the force on each atom is below 0.005 eV Å$^{-1}$. The Perdew–Burke–Ernzerhof (PBE)[52] generalized gradient approximation (GGA) exchange-correlation functional was used throughout. A $(9 \times 9 \times 9)$ Monkhorst-Pack k-grid scheme was used for the calculations of c-CoSe$_2$, o-CoSe$_2$, and m-CoSe$_2$ of $N = 1$; $(9 \times 9 \times 5)$ Monkhorst-Pack k-grid scheme for the m-CoSe$_2$ of $N = 2$. The atomic positions and lattice constants of the pyrite, marcasite, and mixed phases are all optimized. The optimized bulk cell of pyrite is $a = b = c = 5.828$ Å; the optimized lattice constants of marcasite are a = 5.811 Å, b = 4.886 Å, and c = 3.638 Å. Note that the $b$ constant and the [101] length ($\sqrt{a^2 + c^2} = 6.092$ Å) of marcasite are similar to the pyrite lattice constant, with lattice mismatches of 0.3% and 4.5%, respectively.

## Data availability

The data that support the findings of this study are available from the corresponding authors upon request.

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

## Acknowledgements

This work was supported by the funding support from the National Natural Science Foundation of China (Grants 21521001, 21431006, 21225315, 21321002, 91645202, and 51702312, 21975237), the Users with Excellence and Scientific Research Grant of Hefei Science Center of CAS (2015HSCUE007), the Key Research Program of Frontier Sciences, CAS (Grant QYZDJ-SSW-SLH036), the National Basic Research Program of China (Grants 2014CB931800, 2018YFA0702001), the Chinese Academy of Sciences (Grants KGZD-EW-T05, XDA090301001), the Strategic Priority Research Program of the Chinese Academy of Sciences (XDA21000000), the Fundamental Research Funds for the Central Universities (WK2060190045, WK2340000076), and the Recruitment Program of Global Youth Experts. We would like to thank the beamline 1W1B station in the Beijing Synchrotron Radiation Facility and BL14W1 at the Shanghai Synchrotron Radiation Facility for help with the characterizations. This work was partially carried out at the USTC Center for Micro and Nanoscale Research and Fabrication.

## Author contributions

S.-H.Y. and M.-R.G. conceived and supervised the project. X.-L.Z. and Y.-R.Z. performed the experiments, collected, and analyzed the data. X.-S.Z. and J.-F.Z. performed XPS and UPS measurements. C.M. performed the HAADF-STEM measurements. S.-J.H. and X.Z. carried out the DFT calculations. R.W., F.-Y.G., P.-P.Y., Z.-Z.N., C.G. and X.-X.Y. helped with electrochemical data collection and analysis. S.-H.Y., M.-R.G. and X.-L.Z. co-wrote the manuscript. All authors discussed the results and commented on the manuscript.

## Competing interests

The authors declare no competing interests.
