## [Peer Review File · Nature Communications]

Reviewers' comments:

Reviewer #1 (Remarks to the Author):

Yu and coworkers converted c-CoS₂ to polymorphic CoS₂ by controlled thermal processing in alkaline media, and observed markedly enhanced HER performance and stability. The results are interesting and might be publishable in the journal pending the following issues

1. In Fig 2d, there appears to be no difference of the XRD peak width between the m- and c-/o- samples. This suggests comparable crystalline domain size. Yet, since m-CoS₂ is produced by crystal engineering from c-CoS₂, should one expect a smaller domain size in m-CoS₂ than those of c-CoS₂ and o-CoS₂?

A related question, from TEM measurements, the o and c domains are clearly defined and well-separated. Can the authors estimate the typical size of these domains? By the way, the TEM results appear to be at odds with the structural model shown in Fig 5b where the authors argued that these two phases are homogeneously mixed.

2. The authors included rather comprehensive characterization and stated that the m-CoS₂ surface is more stable than the o- and c- counterparts. In general, a more stable surface is not favored for catalysis. There is a sticking point about this study: what exactly is responsible for the enhanced HER activity and durability? The authors argued that the X-ray data suggest enhanced covalency of the Co-S bonds. What exactly does this mean and how is it involved in HER?

3. The authors used nitroso-R to compare the leaching property of the samples. One should note that the m-CoS₂ sample was obtained after a rather harsh, thermochemical treatment, whereas the other two samples were not. That means no unstable, amorphous components would survive in m-CoS₂. Thus, a more meaningful comparison of the stability of the three samples should be done when all samples had been treated in a consistent fashion.

Reviewer #2 (Remarks to the Author):

In the present manuscript, the authors describe a new way to improve the stability of polymorphic cobalt diselenide (CoSe₂) catalyst through homogeneous phase mixing among cubic and orthorhombic phases. A complete set of experiments and material characterization has been performed. Also, XANES spectroscopy and DFT calculations were conducted in order to understand the electronic structure responsible for the stability of the mixed phase catalyst.

The major statement of the manuscript is that the new material retained the catalyst performance for over 400 hours of continuous operation, and it reached stability tests after 50,000 electrochemical cycles. These results may be of relevant importance for the scientific and technological community in the field of catalysis.

In my opinion, the results obtained with this new material are remarkable. However, I consider that some observations should be attended before the manuscript will be published.

Major comments

Page 9 Fig 3 Tafel plots and the analysis should be included here, not in Supplementary Information. The inclusion in the main text of Tafel plots should be accompanied by a deeper discussion about the mechanistic meaning of Tafel slope. A major concern is the way they get the current density which, in the absence of additional information, I suppose is by geometric area.

In Page 9 line 161 to 163 the authors argue that "HER activity of m-CoSe₂ can be ascribed to the large number of nanopores that offer greater accessibility to active sites". For this reason, I consider

that the information regarding the used area—geometric or real- should be included for each electrochemical experiment.

Page 10, lines 176 to 184, the authors describe briefly the impedance analysis performed on the o-, c- and m-CoSe₂.

First, I consider it important to normalize the impedance data by area. Z will have the units Ohm.cm² being the area used the real area not the geometric one. This normalization process by real area will show more reliable impedance values. It will also allow comparing properly the R_{ct} values of the m-CoSe₂ catalyst with the impedance response of the precursor materials o- and c- CoSe₂, that have lower electrochemically active areas.

Second, all the EIS experiments were performed with a polarization at 200 mV. The obtained Nyquist plots show quite different patterns. Fitting with analog electrical circuits should be made in order to identify processes and thus determine R_{ct} values.

Third, it is important to know the j-t behavior at that polarization potential, to ensure whether under these circumstances the measurements were made under pseudo-stationary conditions.

Page 15 line 282, under the title Material Synthesis, the authors describe the synthesis of o-CoSe₂/DETA nanobelts. This material is NOT studied or used in experiments discussed throughout the manuscript.

Minor comments

The frequent reference to Figures and results presented in Supplementary Information makes it very difficult to be read fluently.

Page 7, Figure 2 b, HADDF should be changed to HAADF

We thank all the reviewers for their valuable comments and questions that help us significantly improve the revised manuscript.

REVIEWER REPORTS:

Reviewer #1 (Remarks to the Author):

Yu and coworkers converted c-CoSe₂ to polymorphic CoSe₂ by controlled thermal processing in alkaline media, and observed markedly enhanced HER performance and stability. The results are interesting and might be publishable in the journal pending the following issues.

Response: We greatly appreciate the reviewer's high praise and support on the publication of this work.

1. In Fig 2d, there appears to be no difference of the XRD peak width between the m- and c-/o- samples. This suggests comparable crystalline domain size. Yet, since m-CoSe₂ is produced by crystal engineering from c-CoSe₂, should one expect a smaller domain size in m-CoSe₂ than those of c-CoSe₂ and o-CoSe₂?

Response: We thanks the reviewer for carefully reading the data and the thoughtful comments. Commonly, the Debye-Scherrer formula (*i.e.*, $D = K\lambda/B\cos\theta$) is suitable for the calculation of grain size in the range of 1-100 nm, especially for grain sizes of 40-50 nm. Therefore, the size estimation from Debye-Scherrer formula is not suitable for our c- and o-CoSe₂ samples because they are mono-crystalline belt-like morphology with widths of 500 nm and lengths up to several tens of micrometers (see **Supplementary Figure 1 in the revised SI** and also reference *J. Am. Chem. Soc.* **2009**, 131, 7486).

Figure R1. XRD patterns show the diffraction peaks we selected for determining the FWHM values that used in Debye-Scherrer formula to estimate the domain size.

Following up your comments, we still try to use the XRD diffraction peaks to determine the crystalline domain sizes of the studied materials based on Debye-Scherrer formula. We selected the strongest diffraction peaks for the calculation, which are (210) peak for c-CoSe₂, (111) peak for o-CoSe₂, and the (210)/(111) peaks for m-CoSe₂ (see **Figure R1**). According to the Jade software, we determined the FWHM values and then calculated the domain sizes (see **Table R1**). Results show that the domain sizes of c-CoSe₂ and o-CoSe₂ are about 10.77 nm and 17.49 nm in the mixed CoSe₂ phase, which are smaller than that in the pure c-CoSe₂ and o-CoSe₂ phases.

Table R1. Comparison the domain sizes of different catalysts that calculated from the Debye-Scherrer formula.

Catalyst	(210) _{cubic}			(111) _{orthorhombic}		
	Center	FWHM	Size	Center	FWHM	Size
c-CoSe ₂	33.64°	0.53°	15.45 nm	/	/	/
o-CoSe ₂	/	/	/	34.54°	0.23°	36.55 nm
m-CoSe ₂	33.78°	0.76°	10.77 nm	34.38°	0.47°	17.49 nm

A related question, from TEM measurements, the o and c domains are clearly defined and well-separated. Can the authors estimated the typical size of these domains? By the way, the TEM results appear to be at odds with the structural model shown in Fig 5b where the authors argued that these two phases are homogeneously mixed.

Response: We thank the reviewer for the good question. As demonstrated above, we selected the strongest diffraction peaks for the calculation; they are (210) peak for c-CoSe₂, (111) peak for o-CoSe₂, and the (210)/(111) peaks for m-CoSe₂ (see **Figure R1**). According to the Jade software, we determined the FWHM values and then calculated the domain sizes (see **Table R1**). Our results show that the domain sizes of c-CoSe₂ and o-CoSe₂ are about 10.77 nm and 17.49 nm in the mixed CoSe₂ phase, which are smaller than that in the pure c-CoSe₂ and o-CoSe₂ phases.

We are sorry that **Figure 5b** in our original manuscript is not good to reflect the phase-mixed CoSe₂ material with nearly homogeneous distribution of c- and o-phases. On the basis of your comments, we re-made the structural model and offered it as **Figure 5b in our revised manuscript**.

2. The authors included rather comprehensive characterization and stated that the m-CoSe₂ surface is more stable than the o- and c- counterparts. In general, a more stable surface is not favored for catalysis. There is a sticking point about this study: what exactly is responsible for the enhanced HER activity and durability? The authors argued that the X-ray data suggest enhanced covalency of the Co-Se bonds. What exactly does this mean and how is it involved in HER?

Response: We greatly appreciate the reviewer for his/her thoughtful comments and questions. We totally agree with the reviewer's insight that "In general, a more stable surface is not favored for catalysis", but we would like to note that this is not the sufficient and necessary condition to define a good-or-bad catalyst. Commonly, a more stable surface means that it is reluctant to participate in a reaction. However, this recognition is relative but not absolute. For example, gold (Au) is a very stable metal with dull surfaces that can survive in harsh oxidizing environments, which, on the other hand, can also actively catalyze some reactions, such as alcohol electrooxidation.

In our work, we used X-ray absorption near-edge structure spectroscopy to probe the origin of the observed HER performance of m-CoSe₂ catalyst. Our experimental results uncover a promoted covalency of Co-Se bond in the new phase-mixing material. Greater covalency between Co and Se suggests a decreased occupancy of the antibonding e_g* orbitals of Co 3d, which tends to attract more ligand 4p orbitals. As a result, surface Co ions with high covalency in bonding to Se can boost charge transfer between catalyst surface and adsorbates, leading to higher catalytic activity. We note that the greater covalency that gives superior catalytic properties has also been observed in other catalyst systems such as perovskite oxides with high covalency of transition metal-oxygen bonds (*Science*, **2011**, 334, 1383; *Science*, **2017**, 358, 751). Additionally, this more covalent Co-Se bonding orbital system can strengthen the chemical bonding between Co and Se, which enables substantially improved stability of our m-CoSe₂ catalyst. Similar improved catalytic stability has been previously found in CoS/P/CNT catalyst (*Nat. Commun.* **2016**, 7:10771).

3. The authors used nitroso-R to compare the leaching property of the samples. One should note that the m-CoSe₂ sample was obtained after a rather harsh, thermochemical treatment, whereas the other two samples were not. That means no unstable, amorphous components would survive in m-CoSe₂. Thus, a more meaningful comparison of the stability of the three samples should be done when all samples had been treated in a consistent fashion.

Response: We appreciate the reviewer for this thoughtful suggestion. Following your suggestion, we treated the freshly-prepared c- and o-CoSe₂ in pure DIW at 200 °C for 12h. This process can remove potential unstable and amorphous components existed in the c- and o-CoSe₂ samples, thus offering a fair comparison of the stability of the studied three catalysts.

We thus cycled the three catalysts and their cycled electrolytes were added into the sodium acetate and nitroso-R salt solution. We observed that the green solution turned to light red for c- and o-CoSe₂ samples owing to the formation of Co[C₁₀H₄ONO(SO₃Na)₂]₃ complex (see **Figure R2** below), demonstrating the electrochemical leaching of Co into electrolytes. Our new results again evidences that the superior structural stability of m-CoSe₂ catalyst in acidic solutions.

Figure R2. Colorimetric comparison of the cycled electrolytes added in sodium acetate and nitroso-R salt solution. No color change was seen for $m\text{-CoSe}_2$, indicating its marked structural stability in acidic electrolyte. Nitroso-R salt was used as the color indicator.

Reviewer #2 (Remarks to the Author):

In the present manuscript, the authors describe a new way to improve the stability of polymorphic cobalt diselenide (CoSe_2) catalyst through homogeneous phase mixing among cubic and orthorhombic phases. A complete set of experiments and material characterization has been performed. Also, XANES spectroscopy and DFT calculations were conducted in order to understand the electronic structure responsible of the stability of the mixed phase catalyst.

The major statement of the manuscript is that the new material retained the catalyst performance for over 400 hours of continuous operation, and it reached stability tests after 50,000 electrochemical cycles. These results may be of relevant importance for the scientific and technological community in the field of catalysis.

In my opinion, the results obtained with this new material are remarkable. However, I consider that some observations should be attended before the manuscript will be published.

Response: We greatly appreciate the reviewer for the positive feedbacks on the findings presented in our manuscript.

Major comments

Page 9 Fig 3 Tafel plots and the analysis should be included here, not in Supplementary Information.

The inclusion in the main text of Tafel plots should be accompanied by a deeper discussion about the mechanistic meaning of Tafel slope. A major concern is the way

they get the current density which, in the absence of additional information, I suppose is by geometric area.

Response: We greatly thank the reviewer for his/her experienced comments and suggestion. We accept your suggestion and have moved the Tafel plots to the **revised Main Text**. Further, we have provided suitable discussion on the Tafel analysis, which uncovers a Tafel-step-determined Volmer-Tafel pathway that likely takes effect on the new m-CoSe₂ catalyst (see **the revised MS** for detailed discussion).

Additionally, in our original manuscript, the polarization current densities of studied catalysts were normalized by the geometric surface area. We are sorry for not indicating this clearly in the previous manuscript, which is now properly specified in **the revised MS and SI**.

In Page 9 line 161 to 163 the authors argue that "HER activity of m-CoSe₂ can be ascribed to the large number of nanopores that offer greater accessibility to active sites". For this reason, I consider that the information regarding the used area-geometric or real- should be included for each electrochemical experiment.

Response: We thank the reviewer for the thoughtful comments. In our original manuscript, the polarization current densities of studied catalysts were normalized by the geometric surface area. Although the best way to evaluate the intrinsic electrochemical activity of a catalyst is to calculate its specific activity based on its electrochemically active surface area (ECSA). However, the ECSA values are difficult to obtain for many non-noble metals: they cannot be calculated using the classic hydrogen under-potential deposition (UPD) like commonly done for Pt because no obvious hydrogen adsorption occurs prior to H₂ evolution; they cannot be calculated using capacitance ratio method because the lack of their specific capacitances. As a consequence, researchers commonly compare polarization curves normalized by the geometric surface area instead of the ECSA (we note that the literatures we cited in the manuscript are all based on geometric surface area).

Owing to the above concerns, in our work, we reported the polarization current densities of studied catalysts through normalizing by the geometric surface area. We are sorry for not indicating this clearly in the original manuscript, which is now properly indicated in **the revised MS and SI**.

Page 10, lines 176 to 184, the authors describe briefly the impedance analysis performed on the o-, c- and m-CoSe₂.

First, I consider it important to normalize the impedance data by area. Z will have the units Ohm.cm² being the area used the real area not the geometric one. This

normalization process by real area will show more reliable impedance values. It will also allow comparing properly the R_{ct} values of the m-CoSe₂ catalyst with the impedance response of the precursor materials o- and c- CoSe₂, that have lower electrochemically active areas.

Response: We thank the reviewer for the insightful comments and suggestion. We agree with the reviewer that normalizing the impedance by real area will give a more reliable values and thus a better comparison. However, the electrochemically active surface area (ECSA) values are difficult to obtain for many non-noble metals: they cannot be calculated using the classic hydrogen under-potential deposition (UPD) like commonly done for Pt because no obvious hydrogen adsorption occurs prior to H₂ evolution; they cannot be calculated using capacitance ratio method because the lack of their specific capacitances.

In this case, we estimate the ECSA from the double-layer capacitance according to equation $ECSA = C_{dl}/C_s$ (where C_s is the specific capacitance of the catalyst or the capacitance of smooth planar surface per unit area) that suggested by McCrory, Peters and Jaramillo (*J. Am. Chem. Soc.* 2013, 135, 16977). For the estimate of surface area, we adopted the general specific capacitance of $C_s = 0.035 \text{ mF cm}^{-2}$ (value used in acid) based on typical reported value (*J. Am. Chem. Soc.* 2013, 135, 16977). Accordingly, the ECSA values of various studied catalysts were obtained.

In the **revised MS and SI**, we carefully normalized all the impedance data by the calculated ECSA and all the impedance data in the original MS and SI were properly modified, as suggested by the reviewer.

Second, all the EIS experiments were performed with a polarization at 200 mV. The obtained Nyquist plots show quite different patterns. Fitting with analog electrical circuits should be made in order to identify processes and thus determine R_{tc} values.

Response: We greatly appreciate the reviewer for his/her useful suggestion. Following up your suggestion, we have fitted our EIS data of studied catalysts to the equivalent circuit (see **Figure R3** below). On the basis of these fitting results, we determined that the R_{ct} value of m-CoSe₂ is 1524 Ohm cm_{ECSA}^2 , versus ~13660 Ohm cm_{ECSA}^2 for c-CoSe₂ and ~8100 Ohm cm_{ECSA}^2 for o-CoSe₂. The smallest R_{ct} value of m-CoSe₂ suggests its superior charge transfer kinetics. We further note that no appreciable change for R_{ct} was observed for m-CoSe₂ even after 50,000 cycles, whereas the R_{ct} values substantially increased to 97790 Ohm cm_{ECSA}^2 for c-CoSe₂ and 22651 Ohm cm_{ECSA}^2 for o-CoSe₂ after 2,000 cycles, exhibiting the remarkable catalytic and structural stability of the new m-CoSe₂ catalyst in acid.

Figure R3. a-c, EIS Nyquist plots of the m-, c- and o-CoSe₂ catalysts before and after different potential cycles, respectively. Insets in **a**, **b**, **c** are the corresponding equivalent circuits. Solid lines in **a**, **b**, **c** are the fitting results according to the respective equivalent circuit. **d**, Comparison of the R_{ct} values of the m-, c- and o-CoSe₂ catalysts before and after different potential cycles. R_{sol} and R_{ct} represent the solution resistance and charge transfer resistance, respectively.

We have added these new data in the **revised SI** (as **Supplementary Figure 29**) and modified the discussion properly in our **revised MS**.

Third, it is important to know the j - t behavior at that polarization potential, to ensure whether under these circumstances the measurements were made under pseudo-stationary conditions.

Response: We thank reviewer for the insightful comments and suggestion. According to your suggestion, we have probed the chronoamperometric response ($j \sim t$) recorded on the m-CoSe₂ catalyst at the constant polarization potential of -0.4 V versus Ag/AgCl in 0.5 M H₂SO₄. Our new data shows that no obvious current perturbation was observed over 12 h of continuous operation, as presented in **Figure R4**. This result reveals that our EIS measurements were conducted under pseudostationary conditions.

Figure R4. Chronoamperometric responses ($j \sim t$) recorded on m-CoSe₂ catalyst at a constant applied potential of -0.4 V versus Ag/AgCl in 0.5 M H₂SO₄ for over 12 hours.

We have added this new result in our revised SI (as **Supplementary Fig. 30**) and provided some discussions over there.

Page 15 line 282, under the title Material Synthesis, the authors describe the synthesis of o-CoSe₂/DETA nanobelts. This material is NOT studied or used in experiments discussed throughout the manuscript.

Response: We thank reviewer for carefully reading our manuscript. In this work, we synthesized m-CoSe₂, c-CoSe₂ and o-CoSe₂ catalysts, and compared their electrocatalytic stability in acidic environment. Here, the o-CoSe₂/DETA nanobelts actually stand for the o-CoSe₂. We wrote “o-CoSe₂/DETA nanobelts” in original manuscript because the o-CoSe₂ nanobelts are synthesized using DETA as soft temperature (see the Experimental section). To avoid any potential misunderstanding, we have changed the phrase “o-CoSe₂/DETA” to “o-CoSe₂” in the **revised MS and SI**.

Minor comments

The frequent reference to Figures and results presented in Supplementary Information makes it very difficult to be read fluently.

Response: We thank the reviewer for the careful comment. In order to clearly describe the new phase-mixing CoSe₂ phase and its outstanding electrocatalytic stability in

acidic electrolyte, we intensively applied multiple advanced characterization techniques, accompanied with density functional theory calculations. Such comprehensive material and electrochemical studies produced lots of experimental data that we can not include all of them in the Main Text. As a result, many experimental results were presented as the supplements. In view of your comment, we try the best to restructure these data to make them easier to read.

Page 7, Figure 2 b, HADDF should be changed to HAADF

Response: We thank the reviewer for carefully reading our manuscript and pointing out this mistake. This omit has been corrected in our revised MS.

REVIEWERS' COMMENTS:

Reviewer #1 (Remarks to the Author):

The paper was substantially improved in the revision, and the authors did a good job addressing the issues. It is recommended for acceptance for publication in the journal.

Reviewer #2 (Remarks to the Author):

In the Revised Manuscript as well as in Supplementary Information, the authors addressed all comments given and made all relevant changes. For this reason, I recommend publication.

P. S. The point-to-point answers to the referees' comments

REVIEWERS' COMMENTS:

Reviewer #1 (Remarks to the Author):

The paper was substantially improved in the revision, and the authors did a good job addressing the issues. It is recommended for acceptance for publication in the journal.

Response: We thank the reviewer for strong support on the publication of this work.

Reviewer #2 (Remarks to the Author):

In the Revised Manuscript as well as in Supplementary Information, the authors addressed all comments given and made all relevant changes. For this reason, I recommend publication.

Response: We thank the reviewer for strong support on the publication of this work.